# Intertemporal environmental efficiency assessment in China: A new network-based dynamic super-efficiency measure

**Ruiyue Lin** [1]*, **Zongxin Li** [2]

**1** College of Mathematics and Physics, Wenzhou University, Wenzhou, Zhejiang, PR China, **2** School of Economics and Management, Northwest University, Xi'an, Shaanxi, PR China

* rachel@wzu.edu.cn

**Data Availability Statement:** All relevant data are within the manuscript and its Supporting information files.

**Funding:** Ruiyue Lin recieved the National Natural Science Foundation of China [grant number 71971163]. URL: https://grants.nsfc.gov.cn/

## Abstract

In order to make a complete ranking of intertemporal environmental efficiency in a dynamic manner, this paper combines the network-based dynamic data envelopment analysis (DEA), super-efficiency with the unified efficiency under natural and managerial disposability, and designs a dynamic DEA model and the corresponding dynamic super-efficiency DEA model. Compared with previous studies, the proposed measure can fully rank the overall environmental efficiency of all decision making units (DMUs) in a dynamic manner, and more importantly, it provides the information about when and what factors lead to inefficiency or efficiency of DMUs. The proposed models are applied to examine the environmental efficiency of 30 provinces in China from 2008 to 2017. The results show that there are significant regional differences of environmental efficiency in China. In addition, slack analysis shows that most eastern efficient provinces have no obvious advantages in energy consumption, labor and waste water emission; most central and western efficient provinces have no advantages in sulfur dioxide (SO2) emissions and GDP. To improve overall efficiency, eastern inefficient provinces should mainly focus on reducing energy consumption, SO2 emissions and labor, and increasing capital investment in right years, central and western inefficient provinces can focus on reducing SO2 emissions and labor in most years, most of provinces need to increase gross domestic capital formation.

## Introduction

As a populous country in the world, China consumes a lot of energy when developing its economy. However, it produces a lot of pollutants, which also cause severe environmental problems. In view of these facts, environmental efficiency evaluation methods have been greatly focused and enriched in China in recent years. Data envelopment analysis (DEA) [1], which is a data-driven approach for measuring the relative performance of homogeneous decision making units (DMUs) with multiple inputs and outputs, is one of the most used environmental efficiency evaluation methods [2–12]. Sueyoshi and Goto [13–16] did lots of researches on DEA models for evaluating environmental efficiency and introduced new concepts of natural disposability and managerial disposability for inputs. Goto et al. [17] divided

egrantindex/funcindex/prjsearch-list?locale=zh_CN. The funders had no role in study design, data collection and analysis, decision to publish, or preparation of the manuscript. Other authors received no specific funding for this work.

**Competing interests:** The authors have declared that no competing interests exist.

the input-output indicators into inputs under natural disposability, inputs under managerial disposability, desirable outputs and undesirable outputs, and then constructed a non-radial DEA measure, called the unified efficiency measure of operation and environment under natural and managerial disposability (UENM), to evaluate the unified efficiency of operation and environment.

UENM measures efficiency under static conditions. When several periods with inter-relations are involved, the overall efficiency should be measured in a dynamic manner by taking into account the inter-relationship between consecutive periods [18]. Otherwise, biased efficiency measurements might lead to misleading results [19]. This is because dynamic settings may give rise to seemingly excessive use of resources which are intended to produce beneficial results in future periods. Recently, dynamic DEA considering carryovers has attracted the attention of scholars. Färe and Grosskopf [20] introduced the dynamic aspects of production into the conventional DEA model. They formulated a dynamic DEA model consisting of multiple periods, each of which is connected by storable inputs and carryover outputs. Based on this study, many different types of dynamic DEA models are proposed. See [18, 21–23] for details. However, these dynamic DEA models lack the power to further discriminate efficient DMUs, since all of them have an overall efficiency score of unity. As a result, they cannot provide the further information on which factors the efficient DMUs perform well in each period. Moreover, they are proposed specifically for environmental efficiency assessment and do not consider a complete classification of indicators as UENM does.

Super-efficiency DEA [24] makes it possible to completely rank efficient DMUs. In the super-efficiency model, the evaluated DMU is excluded from the reference set, and thus the input-oriented (output-oriented) super-efficiency model generates super-efficiency scores larger (smaller) than or equal to unity for efficient DMUs so that efficient DMUs can be discriminated. The original super-efficiency model is radial and is infeasible under variable returns to scale (VRS) [25]. Considering that the super-efficiency score generated by the radial super-efficiency model needs to be assisted by observing the resulting slacks to judge whether the evaluated DMU is efficient or weak-efficient, Tone [26] proposed a slacks-based measure (SBM) super-efficiency model, which is non-radial and is feasible under VRS. Based on this, Du et al. [27] proposed a feasible non-radial additive SBM super-efficiency model. Compared with SBM super-efficiency model, it is linear and can be solved directly by computers without any transformation.

To the best of our knowledge, existing studies do not focus on the complete sequencing of intertemporal environmental efficiency of each DMU in a dynamic manner, nor do they explore when and what causes a DMU to be efficient or inefficient in a dynamic system. To fill in the research gaps, we construct a dynamic super-efficiency measure for assessing intertemporal environmental efficiency by combining dynamic DEA, super-efficiency and UENM. Compared with existing studies, the main contributions of this article are as follows:

1. The proposed method is able to discriminate the overall efficiency of all efficient and inefficient DMUs in a dynamic manner.

2. Different from dynamic SBM models, the dynamic models involved in our method are linear and can be easily solved without any transformation.

3. Our method measures the overall unified efficiency of operation and environment by considering inputs under natural and managerial disposability, desirable outputs and undesirable outputs of each period, and carryovers between two periods.

4. Our method can tell us the time when a DMU is inefficient or efficient and what causes it to be inefficient or efficient.

The rest of the paper is structured as follows: Section 2 provides our dynamic model and dynamic super-efficiency model. Section 3 applies the proposed models to assess the intertemporal environmental efficiency of 30 provinces in China from 2008 to 2017, compares the results with those generated by UENM, analyzes the empirical results and provides policies. Section 4 offers conclusions and future researches.

## Methods

### Dynamic unified efficiency under natural and managerial disposability (DUENM)

In this subsection, we will introduce a network-based dynamic DEA model for assessing the overall unified efficiency of operation and environment under natural and managerial disposability. Based on [17, 28], we classify inputs and outputs into four categories: inputs under natural disposability (called ND inputs simply), inputs under managerial disposability (called MD inputs simply), desirable outputs and undesirable outputs.

We assume that there is a panel data with $n$ DMUs ($j = 1, \ldots, n$) from periods 1 to $T$. At each period $t$ ($t = 1, \ldots, T$), every DMU has $m^-$ ND inputs ($i = 1, \ldots, m^-$), $m^+$ MD inputs ($q = 1, \ldots, m^+$), $s$ desirable outputs ($r = 1, \ldots, s$) and $h$ undesirable outputs ($f = 1, \ldots, h$) along with carryovers to the next period $t + 1$. Let $x_{ij}^{-t}$, $x_{qj}^{+t}$, $g_{rj}^{t}$ and $b_{fj}^{t}$ denote ND inputs, MD inputs, desirable outputs and undesirable outputs of DMU $j$ at period $t$ ($t = 1, \ldots, T$), respectively. Inspired by Tone and Tsutsui [21], we consider three categories of links: good, bad and free, and we symbolize them as $z^{good}$, $z^{bad}$ and $z^{free}$. In order to identify them by period $t$, DMU $j$ and item $l$, we employ, for example, the notation $z_{lj}^{good,t}$, ($l = 1, \ldots, c_{good}, j = 1, \ldots, n, t = 1, \ldots, T$) for denoting good links where $c_{good}$ is the number of good links. These are all observed values up to period $T$. Let the symbol $\alpha$ stand for good, bad or free. Denote the initial value of links as $z_{lj}^{\alpha,0}$. For each possible $j$ and $t$, let $X_j^{-t} = (x_{1j}^{-t}, \ldots, x_{m^-j}^{-t})'$, $X_j^{+t} = (x_{1j}^{+t}, \ldots, x_{m^+j}^{+t})'$, $G_j^t = (g_{1j}^t, \ldots, g_{sj}^t)'$, $B_j^t = (b_{1j}^t, \ldots, b_{hj}^t)'$ and $Z_j^{\alpha,t} = (z_{1j}^{\alpha,t}, \ldots, z_{c_\alpha j}^{\alpha,t})'$, $\forall t, j, \alpha$. On the basis of the dynamic production possibility set (PPS) from Tone and Tsutsui [21], we introduce the following dynamic PPS:

$$
\begin{aligned}
P_v^E = \quad & \left\{ (X^{-t}, X^{+t}, G^t, B^t, Z^{\alpha,t}) \,\middle|\, G^t \leq \sum_{j=1}^n G_j^t \lambda_j^t, B^t \geq \sum_{j=1}^n B_j^t \lambda_j^t, \right. \\
& X^{-t} \geq \sum_{j=1}^n X_j^{-t} \lambda_j^t, X^{+t} \leq \sum_{j=1,}^n X_j^{+t} \lambda_j^t, \quad t = 1, \ldots, T; \\
& Z^{good,t} \leq \sum_{j=1}^n Z_j^{good,t} \lambda_j^t, \quad Z^{bad,t-1} \geq \sum_{j=1}^n Z_j^{bad,t-1} \lambda_j^t, \quad t = 1, \ldots, T, \\
& Z^{free,t} : free, \quad t = 0, \ldots, T; \\
& \sum_{j=1}^n Z_j^{\alpha,t} \lambda_j^t = \sum_{j=1}^n Z_j^{\alpha,t} \lambda_j^{t+1}, \quad \forall \alpha, \quad t = 1, \ldots, T-1; \\
& \left. \sum_{j=1}^n \lambda_j^t = 1, \lambda_j^t \geq 0, j = 1, \ldots, n, \quad t = 1, \ldots, T \right\},
\end{aligned}
$$

where $\lambda_j^t$ ($t = 1, \ldots, T$) is the intensity variable for the period $t$, the constraint $\sum_{j=1}^n \lambda_j^t = 1$

corresponds to the VRS constraint in each period $t$. In $P_v^E$, the right side of the inequalities represents the ideal value of the corresponding variable, and the left side represents the corresponding observed value. For variables where bigger is better, such as desirable outputs, MD inputs and $z_{lj}^{good,t}$, the ideal value should be larger than the observed value; for variables where smaller is better, such as ND inputs, undesirable outputs and $z_{lj}^{bad,t-1}$, the ideal value should be smaller than the observed value. Clearly, DMU can handle free links freely. Each free link can be increased or decreased from the observed one [21]. Since $Z^{free,0}$ might be an input of period 1, we consider $Z^{free,t}$ for $t = 0, \ldots, T$, in $P_v^E$. The continuity of links (carryovers) between periods $t$ and $t + 1$ ($t = 1, \ldots, T - 1$) is guaranteed by

$$\sum_{j=1}^{n} Z_j^{\alpha,t}\lambda_j^t = \sum_{j=1}^{n} Z_j^{\alpha,t}\lambda_j^{t+1}, \quad \forall\alpha, \ t = 1, \ldots, T-1. \tag{1}$$

Constraint (1) is critical for the dynamic model since it ensures that the projections of carryovers in the two adjacent periods are consistent. Since we don't take into account period 0 and period $T + 1$, constraint (1) only applies for $t = 1, \ldots, T - 1$.

Different from Tone and Tsutsui [21], we do not consider fixed links here. This is because this type of links would render the dynamic super-efficiency model to be presented later infeasible. Assume that the fixed links $z_{lk}^{fix,t}$ are considered. Then, the dynamic super-efficiency model will have such a constraint set $z_{lk}^{fix,t} = \sum_{j=1,j\neq k}^{n} z_{lj}^{fix,t}\lambda_j^t, \ l = 1, \ldots, c_{fix}, \ t = 0, \ldots, T$, where $c_{fix}$ is the number of fixed links. Under the VRS assumption,

$\min_{j\neq k}\{z_{lj}^{fix,t}\} \leq \sum_{j=1,j\neq k}^{n} z_{lj}^{fix,t}\lambda_j^t \leq \max_{j\neq k}\{z_{lj}^{fix,t}\}, \ \forall j,t$, holds. So, if $z_{lk}^{fix,t} < \min_{j\neq k}\{z_{lj}^{fix,t}\}$ or $z_{lk}^{fix,t} > \max_{j\neq k}\{z_{lj}^{fix,t}\}$ establishes for any $l$ or $t$, the dynamic super-efficiency model will be infeasible.

By referring to [17], we let $\delta = [m^- + m^+ + s + h + 2 * c_{free} + c_{good} + c_{bad}]^{-1}$, where $c_{bad}$ and $c_{free}$ are the number of bad and free links, respectively. Denote

$$R_i^{x-,t} = \delta(\max_j\{x_{ij}^{t-}\} - \min_j\{x_{ij}^{t-}\})^{-1}, \ i = 1, \ldots, m^-$$

$$R_q^{x+,t} = \delta(\max_j\{x_{qj}^{t+}\} - \min_j\{x_{qj}^{t+}\})^{-1}, \ q = 1, \ldots, m^+,$$

$$R_r^{g,t} = \delta(\max_j\{g_{rj}^t\} - \min_j\{g_{rj}^t\})^{-1}, \ r = 1, \ldots, s, \tag{2}$$

$$R_f^{b,t} = \delta(\max_j\{b_{fj}^t\} - \min_j\{b_{fj}^t\})^{-1}, \ f = 1, \ldots, h,$$

$$R_l^{\alpha,t} = \delta(\max_j\{z_{lj}^{\alpha,t}\} - \min_j\{z_{lj}^{\alpha,t}\})^{-1}, \ \forall\alpha, \ l = 1, \ldots, c_\alpha.$$

Based on $P_v^E$, we propose the following network-based dynamic DEA model called the dynamic unified efficiency under natural and managerial disposability (DUENM) model to assess the

overall efficiency level in a dynamic manner.

$$E_k^* = \min \quad \sum_{t=1}^{T} w^t E_k^t$$

$$\text{s.t.} \quad E_k^t = 1 - (\sum_{i=1}^{m^-} R_i^{x-,t} d_i^{x-,t} + \sum_{q=1}^{m^+} R_q^{x+,t} d_q^{x+,t} + \sum_{r=1}^{s} R_r^{g,t} d_r^{g,t} + \sum_{f=1}^{h} R_f^{b,t} d_f^{b,t}$$

$$+ \sum_{l=1}^{c_{free}} R_l^{free,t-1} d_l^{free-,t-1} + \sum_{l=1}^{c_{bad}} R_l^{bad,t-1} d_l^{bad,t-1}$$

$$+ \sum_{l=1}^{c_{free}} R_l^{free,t} d_l^{free+,t} + \sum_{l=1}^{c_{good}} R_l^{good,t} d_l^{good,t}), \quad t = 1, \ldots, T,$$

$$\sum_{j=1}^{n} x_{ij}^{-t} \lambda_j^t = x_{ik}^{-t} - d_i^{x-,t}, \quad i = 1, \ldots, m^-, \quad t = 1, \ldots, T,$$

$$\sum_{j=1}^{n} x_{qj}^{+t} \lambda_j^t = x_{qk}^{+t} + d_q^{x+,t}, \quad q = 1, \ldots, m^+, \quad t = 1, \ldots, T,$$

$$\sum_{j=1}^{n} g_{rj}^t \lambda_j^t = g_{rk}^t + d_r^{g,t}, \quad r = 1, \ldots, s, \quad t = 1, \ldots, T,$$

$$\sum_{j=1}^{n} b_{fj}^t \lambda_j^t = b_{fk}^t - d_f^{b,t}, \quad f = 1, \ldots, h, \quad t = 1, \ldots, T,$$

$$\sum_{j=1}^{n} z_{lj}^{good,t} \lambda_j^t = z_l^{good,t} + d_l^{good,t}, \quad l = 1, \ldots, c_{good}, \quad t = 1, \ldots, T, \qquad (3)$$

$$\sum_{j=1}^{n} z_{lj}^{bad,t-1} \lambda_j^t = z_l^{bad,t-1} - d_l^{bad,t-1}, \quad l = 1, \ldots, c_{bad}, \quad t = 1, \ldots, T,$$

$$\sum_{j=1}^{n} z_{lj}^{free,t} \lambda_j^t = z_{lk}^{free,t} - d_l^{free,t}, \quad l = 1, \ldots, c_{free}, \quad t = 1, \ldots, T,$$

$$\sum_{j=1}^{n} z_{lj}^{free,0} \lambda_j^1 = z_{lk}^{free,0} - d_l^{free,0}, \quad l = 1, \ldots, c_{free},$$

$$d_l^{free,t} = d_l^{free-,t} - d_l^{free+,t}, \quad l = 1, \ldots, c_{free}, \quad t = 0, \ldots, T,$$

$$d_l^{free-,t} \leq M\delta_l^t, \quad l = 1, \ldots, c_{free}, \quad t = 0, \ldots, T,$$

$$d_l^{free+,t} \leq M(1 - \delta_l^t), \quad l = 1, \ldots, c_{free}, \quad t = 0, \ldots, T,$$

$$\sum_{j=1}^{n} z_{lj}^{\alpha,t} \lambda_j^t = \sum_{j=1}^{n} z_{lj}^{\alpha,t} \lambda_j^{t+1}, \quad t = 1, \ldots, T-1, \alpha = good, \; bad, \; free,$$

$$\sum_{j=1}^{n} \lambda_j^t = 1, \quad t = 1, \ldots, T,$$

$$\lambda_j^t, d_i^{x-,t}, d_q^{x+,t}, d_r^{g,t}, d_f^{b,t} \geq 0, \forall i, q, r, f, j, t,$$

$$d_l^{free-,t}, d_l^{free+,t}, d_l^{good,t}, d_l^{bad,t-1} \geq 0, \; d_l^{free,t}, \; free, \; \forall l, t,$$

$$\delta_l^t, 0 - 1.$$

Some explanations about model (3) should be stated here.

- $w^t$ is the weight of the efficiency of period $t$, which is preset by decision makers and satisfies $\sum_{t=1}^{T} w^t = 1$. We set $w^t = \frac{1}{T}$, $t = 1, \ldots, T$, in the empirical analysis of this paper.

- $d_i^{x-,t}$, $d_q^{x+,t}$, $d_r^{g,t}$ and $d_f^{b,t}$ are the inefficiency slacks representing *ND input excesses*, *MD input shortages*, *desirable output shortages*, and *undesirable output excesses* in each period $t$ ($t = 1, \ldots, T$), respectively.

- $d_l^{\alpha,t}$, $\forall t$, is the slack of the link $z_{lj}^{\alpha,t}$, $\forall \alpha$. For each good link, $d_l^{good,t}$, $\forall t$, represents its *shortage*. For each bad link, $d_l^{bad,t}$, $\forall t$, represents its *excess*. All these slacks should be accounted as inefficiency.

- In every period $t$ ($t = 1, \ldots, T$), DMUs use $m^-$ ND inputs, $m^+$ MD inputs as well as $c_{good}$ bad links and $c_{free}$ free links from period $t-1$ to generate $s$ desirable outputs, $h$ undesirable outputs, $c_{bad}$ good links and $c_{free}$ free links. Therefore, according to the UENM in [17], the first constraint set of model (3) is designed to measure the efficiency of DMU $k$ in period $t$. Let $E_k^{t*}$ be the optimal solution variable. Assume that there are two DMUs (called DMU 1 and DMU 2, respectively) satisfy $E_1^{t*} > E_2^{t*}$. Then in period $t$, the average inefficiency slack of DMU 1 is less than that of DMU 2, and thus DMU 1 is more efficient.

- The 2–5th constraint sets of model (3) are used to measure the excess and shortage of inputs and outputs in each period $t$ ($t = 1, \ldots, T$).

- Each good link $z_{lj}^{good,t}$ is treated as an output of period $t$ ($t = 1, \ldots, T$), so we construct the 6th constraint set and consider its output shortfall in $E_k^t$. Similarly, since each bad link $z_{lj}^{bad,t-1}$ is treated as an input of period $t$ ($t = 1, \ldots, T$), we design the 7th constraint set and consider its input excess in $E_k^t$.

- Unlike good and bad links, the free link $z_{lj}^{free,t}$ is either an output of period $t$ ($t = 1, \ldots, T$) or an input of period $t+1$ ($t = 0, \ldots, T-1$). Its value may be increased or decreased from the observed one. As a result, $d_l^{free,t}$ is free in sign in the 8th constraint set, which belongs to either excess or shortage.

- Since we do not consider period 0 in the dynamic system, we use $\sum_{j=1}^{n} z_{lj}^{free,0} \lambda_j^1$ to represent the expect value of the free link $z_{lk}^{free,0}$ in the 9th constraint set, which is treated as an input of period 1.

- According to [21], we express $d_l^{free,t}$ as a difference of two non-negative variables $d_l^{free-,t}$ and $d_l^{free+,t}$ in the 10th constraint, which satisfy

$$d_l^{free-,t} \times d_l^{free+,t} = 0, \quad l = 1, \ldots, c_{free}, t = 0, \ldots, T. \tag{4}$$

Here, $d_l^{free-,t}$ represents the *excessive* $z_{lj}^{free,t}$ and $d_l^{free+,t}$ represents the *shortage* of $z_{lj}^{free,t}$.

- In order to linearize constraint (4), we introduce the 0–1 integer variable $\delta_l^t$ ($l = 1, \ldots, c_{free}, t = 1, \ldots, T-1$) and a big positive number $M$ by referring to [21]. The

11th and 12th constraints, i.e.,

$$d_l^{free-,t} \leq M\delta_l^t, \ \ l = 1, \ldots, c_{free}, t = 0, \ldots, T,$$

$$d_l^{free+,t} \leq M(1 - \delta_l^t), \ \ l = 1, \ldots, c_{free}, t = 0, \ldots, T,$$

(5)

are equivalent to constraint (4).

- The 13th and 14th constraint sets of model (3) are continuity constraint (1) and the VRS constraint, respectively.

- We do not take into account that $d_l^{free-,T}$ and $d_l^{free+,0}$ in $E_k^t$. This is because $d_l^{free-,T}$ represents the excess of the free link $z_{lj}^{free,T}$, which is treated as an input of period $T + 1$, and $d_l^{free+,0}$ represents the shortage of the free link $z_{lj}^{free,0}$, which is treated as an output of period 0. The dynamic system that we consider does not include periods 0 and $T$. So, both of them are not considered.

Denote the optimal solution variables of model (3) as $\lambda_j^{t*}, d_i^{x-,t*}, d_q^{x+,t*}, d_r^{g,t*}, d_f^{b,t*}$, $d_l^{free-,t*}, d_l^{free+,t*}, d_l^{free,t*}, d_l^{good,t*}, d_l^{bad,t*}$. Due to the VRS constraints, it is easy to know that every inefficiency slack (i.e., $d_i^{x-,t*}, d_q^{x+,t*}, d_r^{g,t*}, d_f^{b,t*}, d_l^{free-,t*}, d_l^{free+,t*}, d_l^{good,t*}, d_l^{bad,t*}$) will be less than the data range of the corresponding item, which is defined by its maximum value minus its minimum value. So it is obvious that the value of $E_k^{t*}$ is between 0 and 1 for each period $t$. As a result, $E_k^*$ is also between 0 and 1.

We judge the performance of DMU $k$ based on the following criteria:

(i) DMU $k$ is overall efficient if and only if $E_k^* = 1$; otherwise, it is overall inefficient.

(ii) DMU $k$ is efficient in period $t$ if and only if $E_k^{t*} = 1, t = 1, \ldots, T$; otherwise it is inefficient in period $t$.

Clearly, DMU $k$ is overall efficient if and only if $E_k^{t*} = 1$ for all $t = 1, \ldots, T$. For simplicity, we call $E_k^{t*}$ and $E_k^*$ the DUENM score in period $t$ and the overall DUENM score for DMU $k$, respectively.

## Dynamic unified super-efficiency under natural and managerial disposability (DUSNM)

Model (3) cannot further discriminate overall efficient DMUs, since all of them have an overall efficiency score of unity. By excluding the DMU under evaluation from the reference set, the super-efficiency model [24] can discriminate efficient DMUs. Based on $P_v^E$, we define the following dynamic super-efficiency PPS for a target DMU $k$ ($k \in \{1, \ldots, n\}$).

$$
\begin{aligned}
P_v^S &= \{(X^{-t}, X^{+t}, G^t, B^t, Z^{\alpha,t}) | G^t \leq \sum_{j=1, j \neq k}^n G_j^t \lambda_j, B^t \geq \sum_{j=1, j \neq k}^n B_j^t \lambda_j, \\
&\quad X^{-t} \geq \sum_{j=1, j \neq k}^n X_j^{-t} \lambda_j, X^{+t} \leq \sum_{j=1, j \neq k}^n X_j^{+t} \lambda_j, \ t = 1, \ldots, T; \\
&\quad \sum_{j=1, j \neq k}^n Z_j^{\alpha,t} \lambda_j^t = \sum_{j=1, j \neq k}^n Z_j^{\alpha,t} \lambda_j^{t+1}, \ \forall \alpha, \ t = 1, \ldots, T-1; \\
&\quad Z^{good,t} \geq \sum_{j=1, j \neq k}^n Z_j^{good,t} \lambda_j^t, \ Z^{bad,t} \leq \sum_{j=1, j \neq k}^n Z_j^{bad,t} \lambda_j^t, \\
&\quad Z^{free,t} : free, \ t = 0, \ldots, T; \\
&\quad \sum_{j=1, j \neq k}^n \lambda_j^t = 1, \ \lambda_j^t \geq 0, j = 1, \ldots, n, j \neq k, \ t = 1, \ldots, T\},
\end{aligned}
$$

where $\lambda_j^t$ ($t = 1, \ldots, T$) is still the intensity variable in period $t$ and the constraint $\sum_{j=1,j\neq k}^n \lambda_j^t = 1$ corresponds to the VRS constraint with respect to super-efficiency in period $t$. Compared with $P_v^E$, $\lambda_k$, which is related to the evaluated DMU $k$, is excluded from $P_v^S$ to achieve the purpose of super-efficiency evaluation.

Non-radial VRS super-efficiency models, such as Super SBM [26] and super additive models [27, 29], are feasible under a positive data set. The DUENM model (i.e., model (3)) is an additive DEA model, and thus is non-radial. Then based on model (3), the dynamic super-efficiency PPS $P_v^S$ and the additive super-efficiency model [27], we propose the following dynamic unified super-efficiency under natural and managerial disposability (DUSNM) model to identify overall efficient DMUs.

$$\bar{E}_k^* = \min \quad \sum_{t=1}^T w^t \bar{E}_k^t$$

$$\text{s.t.} \quad \bar{E}_k^t = 1 + \left(\sum_{i=1}^{m^-} R_i^{x-,t}\tau_i^{x-,t} + \sum_{q=1}^{m^+} R_q^{x+,t}\tau_q^{x+,t} + \sum_{r=1}^s R_r^{g,t}\tau_r^{g,t} + \sum_{f=1}^h R_f^{b,t}\tau_f^{b,t}\right.$$

$$\left. + \sum_{l=1}^{c_{bad}} R_l^{bad,t-1}\tau_l^{bad,t-1} + \sum_{l=1}^{c_{bad}} R_l^{good,t}\tau_l^{good,t}\right), \quad t = 1, \ldots, T,$$

$$\sum_{j=1,j\neq k}^n x_{ij}^{-t}\lambda_j^t \leq x_{ik}^{-t} + \tau_i^{x-,t}, \quad i = 1, \ldots, m^-, \quad t = 1, \ldots, T,$$

$$\sum_{j=1,j\neq k}^n x_{qj}^{+t}\lambda_j^t \geq x_{qk}^{+t} - \tau_q^{x+,t}, \quad q = 1, \ldots, m^+, \quad t = 1, \ldots, T,$$

$$\sum_{j=1,j\neq k}^n g_{rj}^t\lambda_j^t \geq g_{rk}^t - \tau_r^{g,t}, \quad r = 1, \ldots, s, \quad t = 1, \ldots, T,$$

$$\sum_{j=1,j\neq k}^n b_{fj}^t\lambda_j^t \leq b_{fk}^t + \tau_f^{b,t}, \quad f = 1, \ldots, h, \quad t = 1, \ldots, T, \tag{6}$$

$$\sum_{j=1,j\neq k}^n z_{lj}^{good,t}\lambda_j^t \geq z_l^{good,t} - \tau_l^{good,t}, \quad l = 1, \ldots, c_{good}, \quad t = 1, \ldots, T,$$

$$\sum_{j=1,j\neq k}^n z_{lj}^{bad,t-1}\lambda_j^t \leq z_l^{bad,t-1} + \tau_l^{bad,t-1}, \quad l = 1, \ldots, c_{bad}, \quad t = 1, \ldots, T,$$

$$\sum_{j=1,j\neq k}^n z_{lj}^{\alpha,t}\lambda_j^t = \sum_{j=1,j\neq k}^n z_{lj}^{\alpha,t}\lambda_j^{t+1}, \quad t = 1, \ldots, T-1, \quad \alpha = good, bad, free,$$

$$\sum_{j=1,j\neq k}^n \lambda_j^t = 1, \quad t = 1, \ldots, T,$$

$$\lambda_j^t, \tau_i^{x-,t}, \tau_q^{x+,t}, \tau_r^{g,t}, \tau_f^{b,t}, \tau_l^{good,t}, \tau_l^{bad,t-1} \geq 0, \forall i, q, r, f, l, j, t.$$

Some explanations about model (6) are provided here:

- By referring to [27, 29], we use the first constraint set to measure the super-efficiency of DMU $k$ in period $t$. Note that super-efficiency slacks of free links do not be accounted in the first constraint set of model (6). This is because the shortages (excesses) of free links as outputs in period $t$ can also be treated as the savings (surpluses) of the free links as inputs in period $t + 1$. Because we have considered all the positive and negative slacks (i.e., excesses

and shortages) of free links in model (3), taking into account them in the super-efficiency model (6) again becomes repetitive. Therefore, super-efficiency slacks of free links do not appear in this model anymore. This will not affect the ability of our super-efficiency model to identify overall efficient DMUs. They can be well distinguished by savings or surplus of other indicators.

- The 2–7th constraint sets are used to measure the saving and surplus of inputs, outputs and links, respectively.

- $\tau_i^{x-,t}$ and $\tau_q^{x+,t}$ in the second and third constraint sets are super-efficiency slacks representing ND input saving and MD input surplus, respectively.

- $\tau_r^{g,t}$ and $\tau_f^{b,t}$ in the 4th and 5th constraint sets are super-efficiency slacks representing desirable output surplus and undesirable output saving, respectively.

- $\tau_l^{good,t}$ and $\tau_l^{bad,t}$ in the 6th and 7th constraint sets are super-efficiency slacks representing surplus and saving of good and bad links, respectively.

- The 8th constraint set ensures the continuity of links between periods with respect to the super-efficiency.

Let $\bar{E}_k^{t*}, \tau_i^{x-,t*}, \tau_q^{x+,t*}, \tau_r^{g,t*}, \tau_f^{b,t*}, \tau_l^{good,t*}, \tau_l^{bad,t*}$ be the optimal solution variables of model (6). $\tau_i^{x-,t*} > 0$ means that DMU $k$ saves the $i$th ND input than other DMUs does in period $t$. Similar explanations apply to other super-efficiency slacks. It is not difficult to obtain the following cases for the target DMU $k$:

(i) if DMU $k$ lies in $P_v^S$, then all the super-efficiency slacks (i.e., $\tau_i^{x-,t*}, \tau_q^{x+,t*}, \tau_r^{g,t*}, \tau_f^{b,t*}$,

$\tau_l^{good,t*}, \tau_l^{bad,t*}$) in every period $t$ are equal to zero, and thus $\bar{E}_k^{t*} = \bar{E}_k^* = 1, \ \forall t$;

(ii) if DMU $k$ is outside $P_v^S$, then there exists at least one period $t$, in which at least one super-efficiency slack of DMU $k$ is greater than zero and $\bar{E}_k^{t*} > 1$, and thus $\bar{E}_k^* > 1$.

From the constraints of model (6), we know that $\bar{E}_k^*, \bar{E}_k^{t*} \geq 1$. The larger the value of $\bar{E}_k^*$, the larger the savings and surpluses of inputs and outputs, which means the higher the efficiency. We define the period DUSNM score and the overall DUSNM score for each target DMU $k$ as follows:

$$\rho_k^{t*} = \begin{cases} E_k^{t*}, E_k^* < 1, t = 1, \ldots, T, \\ \bar{E}_k^{t*}, E_k^* = 1, t = 1, \ldots, T. \end{cases} \tag{7}$$

$$\rho_k^* = \begin{cases} E_k^*, E_k^* < 1, \\ \bar{E}_k^*, E_k^* = 1. \end{cases} \tag{8}$$

We can see from (8) that $\rho_k^* \geq 1$ if and only if $E_k^* = 1$. Also, $\rho_k^* < 1$ if and only if $E_k^* < 1$. Therefore, DMU $k$ is overall efficient if and only if $\rho_k^* \geq 1$; otherwise, DMU $k$ is overall inefficient. The larger the value of $\rho_k^*$, the more overall efficient the DMU $k$. Therefore, we can judge from the value of $\rho_k^*$ whether the DMU $k$ is efficient in all periods, and we can fully rank all the DMUs according to $\rho_k^*$.

From (7), we can also get the similar conclusion for $\rho_k^{t*}$. In the dynamic situation, we distinguish the efficiency of DMU $k$ in period $t$ according to the following criteria: DMU $k$ is efficient in period $t$ if and only if $\rho_k^{t*} \geq 1$; otherwise, it is inefficient in period $t$. The larger the

value of $\rho_k^{t*}$, the more efficient the DMU $k$ in period $t$. Clearly, $\rho_k^* \geq 1$ if and only if $\rho_k^{t*} \geq 1, \ \forall t$; if $\rho_k^* < 1$, then all the $\rho_k^{t*} \leq 1$ and there exists at least one $\rho_k^{t*} < 1$. Moreover, from (7) and (8) and models (3) and (6), it is easy to get

$$\rho_k^* = \sum_{t=1}^{T} w^t \rho_k^{t*}. \tag{9}$$

Therefore, the overall DUSNM score is a weighted average of period DUSNM scores. If DMU $k$ is overall inefficient, then according to the period DUSNM score $\rho_k^{t*}$ and the slacks generated by model (3), we can know what factors lead to the overall inefficiency of DMU $k$ in which period. If DMU $k$ is overall efficient, then through $\rho_k^{t*}$ and the super-efficiency slacks generated by model (6), we also can know which factor of DMU $k$ is more advantageous than other DMUs in which period.

## Empirical research

In this section, we use our method to measure the intertemporal environmental efficiency of different provinces in China so as to show the effectiveness of our new models and meanwhile analyze the efficiency of each provinces.

### Data sources and indicators

In this paper, 30 provinces or province-equivalents (called provinces for short, hereafter) in China are taken as DMUs. This is due to the lack of energy data in Tibet and the differences in statistical data methods between Hong Kong, Macao and other provinces in mainland China, and thus they are excluded from our data set. The time horizon is from 2008 to 2017 and we treat a calendar year as one period.

According to the standard of China's economic area, 30 sample provinces are grouped into three major areas to facilitate the comparison. Table 1 lists the detailed grouping information. Such grouping is widely used in environmental efficiency assessment research [30].

Goto et al. [17] considered labor and energy as two ND inputs, capital as the unique MD outputs. Considering that fixed assets investment of each province can reflect the investment and utilization of capital, we use fixed asset investment to represent capital. According to [31], we use employed population at the end of the year to represent labor and annual energy consumption to represent energy. Existing studies, such as [30, 31], usually adopt economic indicators as desirable outputs and pollutant emissions as undesirable outputs. We take the gross domestic product (GDP) as the unique desirable output, which can reflect the level of economic development and overall strength of China's provinces. SO2 emission is a main pollutant in the exhaust gas and waste water emission is the main source of water pollution. Therefore, we treat them as undesirable outputs. We adopt the gross domestic capital formation as the free carryover variable. This index can not only represent the output of the current period, but also is the input that the next period gets from the previous period.

**Table 1. The detailed grouping information of 30 provinces.**

| Area | Province |
|------|----------|
| Eastern | Beijing, Tianjin, Shanghai, Shandong, Hebei, Jiangsu, Zhejiang, Fujian, Guangdong, Hainan, Liaoning |
| Central | Jilin, Heilongjiang, Henan, Shanxi, Anhui, Hubei, Hunan, Jiangxi |
| Western | Chongqing, Sichuan, Guizhou, Yunnan, Guangxi, Shaanxi, Gansu, Qinghai, Inner Mongolia, Ningxia, Xinjiang |

**Table 2. Descriptive statistics.**

| Variables | Indicators | Source | Max | Min | Stdev | Ave |
|---|---|---|---|---|---|---|
| ND inputs | Energy consumption (Ten thousand tons standard coal) | China Energy Statistical Yearbook | 1768137 | 1135.33 | 101627.56 | 19470.72 |
| | Employed population (Ten thousand) | National Bureau of Statistics of China | 6766.86 | 301 | 1745.16 | 2666.43 |
| MD Input | Fixed assets investment (100 million) | China Economic and Social Big Data Research Platform | 55202.72 | 583.24 | 10579.22 | 13548.30 |
| Desirable output | GDP (100 million) | National Bureau of Statistics of China | 89705.23 | 1018.62 | 16242.88 | 19648.17 |
| Undesirable output | SO2 emissions (Ten thousand tons) | National Bureau of Statistics of China | 1827397.20 | 14271.49 | 407211.93 | 630274.00 |
| | Waste water emissions (Ten thousand tons) | National Bureau of Statistics of China | 938261.03 | 19997 | 175049.87 | 222490.26 |
| Carray-over | Gross domestic capital formation (100 million) | China Economic and Social Big Data Research Platform | 39657.52 | 496.71 | 7331.83 | 10439.13 |

Table 2 shows data sources and the descriptive statistics of the data. We can find the following facts from the maximum and minimum values for each indicator: (A) Energy consumption: Inner Mongolia has the maximum in 2013, while Hainan has the minimum in 2008. (B) Fixed assets investment: Shandong has the maximum in 2017, while Qinghai has the minimum in 2008. (C) Employed population: Henan achieves the maximum in 2017, while Qinghai achieves the minimum in 2008. (D) SO2 emissions: Shandong has the maximum in 2011, while Hainan has the minimum in 2017. (E) GDP: Guangdong has the maximum in 2017, while Qinghai has the minimum in 2008. (F) Waste water emissions and gross domestic capital formation: Guangdong has the maximum in 2017, while Qinghai has the minimum in 2007.

## Comparison

We use matlab 2020a to solve all the related linear programmes and the detailed results are shown in Table 3. Since DEA is a dimensionless technology, all the indicator data are directly substituted into the models to calculate without data processing. Columns 3–12 of Table 3 list the period DUSNM scores of each province in every sample year. The overall DUSNM scores and the rankings are provided in Columns 13 and 14 of Table 3. The values of $E_k^*$ and $\bar{E}_k^*$, which are generated by models (3) and (6), are provided in Columns 15 and 16. For comparison, we also apply the UENM model [17] to this data set. Since the UENM model is static, we apply the UENM model to each year separately. In each period, we not only consider the inputs and outputs of this period, but regard the carryover of the previous period as an input and the carryover of this period as an output. The last column of Table 3 shows the annual average UENM score for each province.

It is not hard to see that there are 12 provinces with an annual average UENM equal to 1. This means that annual average UENM scores cannot distinguish the efficiency of the 12 provinces. Similarly, there are 12 provinces with $E_k^*$ equal to 1 and 18 provinces with $\bar{E}_k^*$ equal to 1. The values of $E_k^*$ cannot ranks overall efficient provinces and those of $\bar{E}_k^*$ cannot ranks overall inefficient provinces. So, we cannot rank the performance of all the provinces just according to $E_k^*$ or $\bar{E}_k^*$. The overall DUSNM scores can solve this issue. Column 14 of Table 3 provides the full ranking of provinces. We see that Jiangsu has the highest overall DUSNM score, while Shanxi's overall DUSNM score is at the bottom. Twelve overall efficient provinces rank according to their overall DUSNM scores as follows:

Jiangsu≻Guangdong≻Shandong≻Beijing≻Hainan≻Tianjin≻Anhui≻Qinghai≻Inner Mongolia ≻Shanghai≻Jiangxi≻Chongqing, where the symbol "≻" represents "performs better than".

**Table 3. DUSENM and UENM scores.**

| Area | Province | $\rho_k^{t*}$ | | | | | | | | | | $\rho_k^*$ | rank | $E_k^*$ | $\bar{E}_k^*$ | Average of UENM |
|---|---|---|---|---|---|---|---|---|---|---|---|---|---|---|---|---|
| | | 2008 | 2009 | 2010 | 2011 | 2012 | 2013 | 2014 | 2015 | 2016 | 2017 | | | | | |
| Eastern | Beijing | 1.0225 | 1.0216 | 1.0219 | 1.0201 | 1.0205 | 1.0162 | 1.0207 | 1.0211 | 1.0177 | 1.0173 | 1.0200 | 4 | 1.0000 | 1.0200 | 1.0000 |
| Eastern | Tianjin | 1.0114 | 1.0166 | 1.0170 | 1.0187 | 1.0140 | 1.0138 | 1.0165 | 1.0165 | 1.0195 | 1.0128 | 1.0157 | 6 | 1.0000 | 1.0157 | 1.0000 |
| Eastern | Hebei | 0.8805 | 0.9291 | 0.9115 | 0.8951 | 0.9029 | 0.9319 | 0.9354 | 1.0000 | 0.9879 | 1.0000 | 0.9374 | 22 | 0.9374 | 1.0003 | 0.9605 |
| Eastern | Liaoning | 1.0000 | 1.0000 | 1.0000 | 1.0000 | 1.0000 | 1.0000 | 0.9891 | 0.9539 | 0.8814 | 0.8436 | 0.9668 | 20 | 0.9668 | 1.0122 | 0.9602 |
| Eastern | Shanghai | 1.0116 | 1.0084 | 1.0073 | 1.0061 | 1.0045 | 1.0000 | 1.0000 | 1.0010 | 1.0006 | 1.0039 | 1.0043 | 10 | 1.0000 | 1.0043 | 1.0000 |
| Eastern | Jiangsu | 1.0423 | 1.0492 | 1.0534 | 1.0595 | 1.0623 | 1.0675 | 1.0694 | 1.0687 | 1.0629 | 1.0659 | 1.0601 | 1 | 1.0000 | 1.0601 | 1.0000 |
| Eastern | Zhejiang | 0.9909 | 0.9722 | 0.9673 | 1.0000 | 1.0000 | 1.0000 | 1.0000 | 1.0000 | 1.0000 | 0.9953 | 0.9926 | 13 | 0.9926 | 1.0004 | 0.9894 |
| Eastern | Fujian | 0.9470 | 0.9479 | 0.9581 | 0.9489 | 1.0000 | 1.0000 | 1.0000 | 1.0000 | 0.9923 | 1.0000 | 0.9794 | 17 | 0.9794 | 1.0025 | 0.9904 |
| Eastern | Shandong | 1.0313 | 1.0259 | 1.0211 | 1.0242 | 1.0194 | 1.0159 | 1.0143 | 1.0145 | 1.0233 | 1.0153 | 1.0205 | 3 | 1.0000 | 1.0205 | 1.0000 |
| Eastern | Guangdong | 1.0233 | 1.0197 | 1.0175 | 1.0261 | 1.0255 | 1.0205 | 1.0201 | 1.0189 | 1.0309 | 1.0292 | 1.0232 | 2 | 1.0000 | 1.0232 | 1.0000 |
| Eastern | Hainan | 1.0152 | 1.0154 | 1.0160 | 1.0161 | 1.0167 | 1.0094 | 1.0182 | 1.0177 | 1.0181 | 1.0206 | 1.0163 | 5 | 1.0000 | 1.0163 | 1.0000 |
| Central | Shanxi | 0.8436 | 0.8633 | 0.8587 | 0.8591 | 0.8645 | 0.8891 | 0.8468 | 0.8518 | 0.8582 | 0.8135 | 0.8549 | 30 | 0.8549 | 1.0000 | 0.8680 |
| Central | Jilin | 1.0000 | 1.0000 | 0.9989 | 0.9751 | 0.9832 | 0.9969 | 1.0000 | 1.0000 | 1.0000 | 0.9691 | 0.9923 | 14 | 0.9923 | 1.0009 | 0.9869 |
| Central | Heilongjiang | 0.9205 | 0.9294 | 0.9322 | 0.9266 | 0.9359 | 0.9506 | 0.9287 | 0.9223 | 0.9108 | 0.9066 | 0.9264 | 23 | 0.9264 | 1.0000 | 0.9456 |
| Central | Anhui | 1.0061 | 1.0060 | 1.0087 | 1.0061 | 1.0080 | 1.0078 | 1.0090 | 1.0065 | 1.0070 | 1.0045 | 1.0070 | 7 | 1.0000 | 1.0070 | 1.0000 |
| Central | Jiangxi | 1.0015 | 1.0035 | 1.0050 | 1.0036 | 1.0038 | 1.0001 | 1.0029 | 1.0029 | 1.0033 | 1.0054 | 1.0032 | 11 | 1.0000 | 1.0032 | 1.0000 |
| Central | Henan | 0.8890 | 0.8860 | 1.0000 | 1.0000 | 1.0000 | 1.0000 | 1.0000 | 1.0000 | 0.9869 | 1.0000 | 0.9762 | 18 | 0.9762 | 1.0018 | 1.0000 |
| Central | Hubei | 0.8866 | 0.8967 | 0.9086 | 0.9117 | 0.9261 | 0.9546 | 0.9640 | 1.0000 | 0.9761 | 1.0000 | 0.9424 | 21 | 0.9424 | 1.0008 | 0.9736 |
| Central | Hunan | 0.8700 | 0.8771 | 0.8842 | 0.9015 | 0.9103 | 0.9198 | 0.9285 | 0.9446 | 0.9413 | 0.9710 | 0.9148 | 25 | 0.9148 | 1.0000 | 0.9617 |
| Western | Inner Mongolia | 1.0080 | 1.0089 | 1.0033 | 1.0034 | 1.0039 | 1.0037 | 1.0120 | 1.0044 | 1.0010 | 1.0008 | 1.0049 | 9 | 1.0000 | 1.0049 | 1.0000 |
| Western | Guangxi | 0.8413 | 0.8585 | 0.8563 | 0.9075 | 0.9382 | 0.9493 | 0.9525 | 0.9555 | 0.9523 | 0.9456 | 0.9157 | 24 | 0.9157 | 1.0000 | 0.9650 |
| Western | Chongqing | 1.0016 | 1.0000 | 1.0000 | 1.0000 | 1.0000 | 1.0000 | 1.0002 | 1.0010 | 1.0007 | 1.0024 | 1.0006 | 12 | 1.0000 | 1.0006 | 0.9922 |
| Western | Sichuan | 0.8424 | 0.8808 | 0.8636 | 0.8736 | 0.8874 | 0.8977 | 0.8830 | 0.8869 | 0.8838 | 0.8876 | 0.8787 | 28 | 0.8787 | 1.0000 | 0.9376 |
| Western | Guizhou | 0.8537 | 0.8484 | 0.8520 | 0.8660 | 0.8730 | 0.8932 | 0.8883 | 0.9024 | 0.9039 | 0.8851 | 0.8766 | 29 | 0.8766 | 1.0000 | 0.9457 |
| Western | Yunnan | 0.9072 | 0.9119 | 0.9122 | 0.8932 | 0.8975 | 0.9095 | 0.9004 | 0.9074 | 0.8860 | 0.9177 | 0.9043 | 27 | 0.9043 | 1.0000 | 0.9849 |
| Western | Shaanxi | 0.9285 | 0.9448 | 0.9453 | 1.0000 | 1.0000 | 1.0000 | 1.0000 | 1.0000 | 0.9953 | 1.0000 | 0.9814 | 15 | 0.9814 | 1.0030 | 0.9806 |
| Western | Gansu | 0.9321 | 0.9845 | 0.9847 | 0.9846 | 0.9856 | 0.9863 | 0.9843 | 0.9823 | 0.9833 | 0.9076 | 0.9715 | 19 | 0.9715 | 1.0001 | 0.9808 |
| Western | Qinghai | 1.0047 | 1.0049 | 1.0046 | 1.0049 | 1.0050 | 1.0052 | 1.0055 | 1.0051 | 1.0049 | 1.0052 | 1.0050 | 8 | 1.0000 | 1.0050 | 1.0000 |
| Western | Ningxia | 0.9961 | 0.9935 | 0.9952 | 0.9963 | 0.9709 | 0.9755 | 0.9687 | 0.9663 | 0.9746 | 0.9637 | 0.9801 | 16 | 0.9801 | 1.0001 | 0.9879 |
| Western | Xinjiang | 0.9210 | 0.9155 | 0.9152 | 0.9115 | 0.9097 | 0.9221 | 0.8962 | 0.8948 | 0.8900 | 0.8890 | 0.9065 | 26 | 0.9065 | 1.0000 | 0.9571 |
| | Average | 0.9543 | 0.9607 | 0.9640 | 0.9680 | 0.9723 | 0.9779 | 0.9752 | 0.9782 | 0.9731 | 0.9693 | 0.9693 | | 0.9633 | 1.0068 | 0.9789 |

In addition to the advantage of providing a complete ranking, our method can tell us when overall inefficient provinces perform poorly. For example, Hebei and Hubei are efficient only in 2015 and 2017, but are inefficient in other sample years. Notice that ten provinces are inefficient in each sample year. They are Shanxi, Heilongjiang, Hunan, Guangxi, Sichuan, Guizhou, Yunnan, Gansu, Ningxia and Xinjiang.

## Performance analysis

The spatial distribution of overall DUSNM scores for the 30 provinces is shown in Fig 1, where the color warmth represents the overall DUSNM level. As can be seen from Fig 1, most overall efficient provinces are in the eastern area and the bottom five provinces in overall DUSNM scores are all in central and west areas.

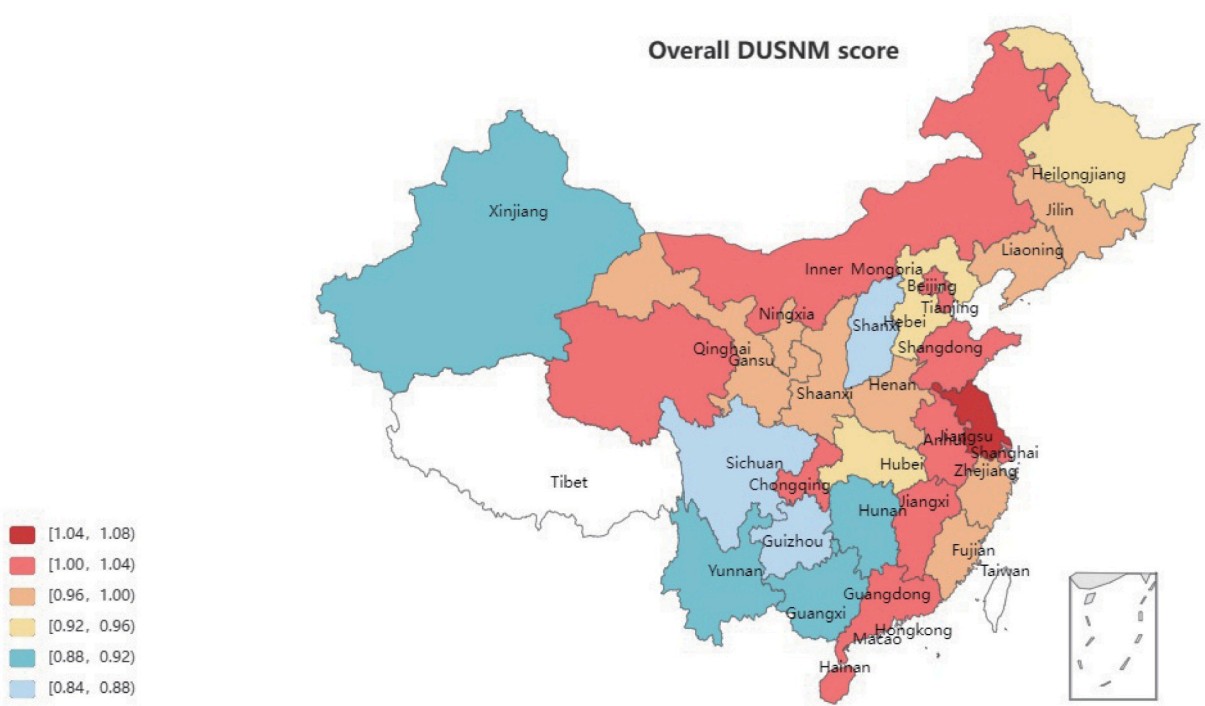

**Fig 1. Spatial distribution of overall DUSNM scores for the 30 provinces in China.** The color warmth represents the overall DUSNM scores of 30 provinces. The warmer the color, the higher the overall DUSNM score.

Fig 2 shows the ranking curves about the period DUSNM score $\rho_k^{t*}$ of eastern, central and western areas, respectively. From Fig 2, we find that there are some provinces with high ranking fluctuations. For example, Henan jumps to the 13th place in 2010, and its ranking falls back in 2016, while its ranking from 2011 to 2017 is still better than that of 2008 and 2009. It is not difficult to see from Fig 2: (i) in the eastern area, Hebei, Liaoning, Zhejiang and Fujian rank relatively low; (ii) in the western area, Qinghai and Inner Mongolia rank relatively high, but Xinjiang, Yunnan, Sichuan and Guizhou rank very low each year; (iii) in the central area, Shanxi, Heilongjiang and Hunan perform poorly in each sample year.

The period DUSNM scores of provinces show an unbalanced situation, with obvious area differences. Table 4 shows the descriptive statistics of average DUSNM scores of three areas in each period. We can see that in each year, the average and maximum DUSNM scores of all the provinces in the eastern area are larger than those in the central and western areas, while the minimum DUSNM score of all provinces in the eastern area is larger than that in the central and western areas in most sample years. Except in 2008, 2009 and 2011, the average DUSNM score of all the provinces in the central area is constantly higher than that in the western area.

This shows that the efficiency of eastern provinces is better than that of central and western provinces. Moreover, in most years, the efficiency of central provinces is higher than that of western provinces.

## Slack analysis

One advantage of our method is that based on the values of slacks generated by models (3) and (6), we can tell exactly when and what causes these overall efficient (inefficient) provinces to be overall efficient (inefficient). Table 5 gives the dimensionless super-efficiency slacks of the

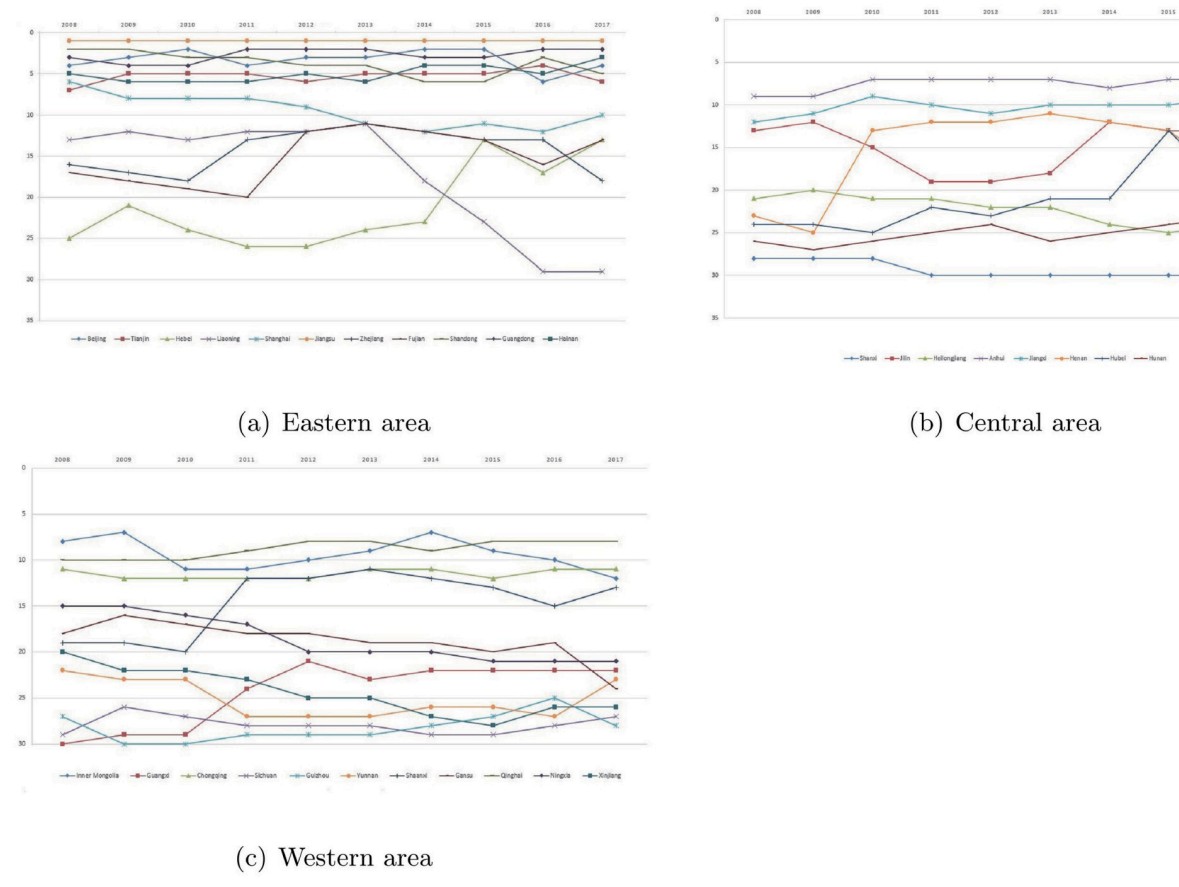

(a) Eastern area

(b) Central area

(c) Western area

**Fig 2. Period DUSNM ranking curves.** The ranking curves show the ranking of the period DUSNM score $\rho_k^{t*}$ of provinces in the eastern, central, western areas, respectively, which can be calculated from the data of $\rho_k^{t*}$ in Table 3. The vertical axis is the rank and the horizontal axis is the year. The smaller the ranking, the larger the $\rho_k^{t*}$.

**Table 4. Descriptive statistics of annual average DUSNM scores of three areas.**

| Year | | 2008 | 2009 | 2010 | 2011 | 2012 | 2013 | 2014 | 2015 | 2016 | 2017 |
|---|---|---|---|---|---|---|---|---|---|---|---|
| Eastern | Avg. | 0.9978 | 1.0005 | 0.9992 | 1.0013 | 1.0060 | 1.0068 | 1.0076 | 1.0102 | 1.0032 | 1.0004 |
| | Max | 1.0423 | 1.0492 | 1.0534 | 1.0595 | 1.0623 | 1.0675 | 1.0694 | 1.0687 | 1.0629 | 1.0659 |
| | Min | 0.8805 | 0.9291 | 0.9115 | 0.8951 | 0.9029 | 0.9319 | 0.9354 | 0.9539 | 0.8814 | 0.8436 |
| | S.D. | 0.0441 | 0.0344 | 0.0373 | 0.0421 | 0.0367 | 0.0300 | 0.0303 | 0.0258 | 0.0433 | 0.0529 |
| Central | Avg. | 0.9272 | 0.9327 | 0.9495 | 0.9480 | 0.9540 | 0.9649 | 0.9600 | 0.9660 | 0.9604 | 0.9588 |
| | Max | 1.0061 | 1.0060 | 1.0087 | 1.0061 | 1.0080 | 1.0078 | 1.0090 | 1.0065 | 1.0070 | 1.0054 |
| | Min | 0.8436 | 0.8633 | 0.8587 | 0.8591 | 0.8645 | 0.8891 | 0.8468 | 0.8518 | 0.8582 | 0.8135 |
| | S.D. | 0.0617 | 0.0574 | 0.0571 | 0.0522 | 0.0492 | 0.0410 | 0.0527 | 0.0523 | 0.0497 | 0.0630 |
| Western | Avg. | 0.9306 | 0.9411 | 0.9393 | 0.9492 | 0.9519 | 0.9584 | 0.9537 | 0.9551 | 0.9524 | 0.9459 |
| | Max | 1.0080 | 1.0089 | 1.0046 | 1.0049 | 1.0050 | 1.0052 | 1.0120 | 1.0051 | 1.0049 | 1.0052 |
| | Min | 0.8413 | 0.8484 | 0.8520 | 0.8660 | 0.8730 | 0.8932 | 0.8830 | 0.8869 | 0.8838 | 0.8851 |
| | S.D. | 0.0626 | 0.0583 | 0.0594 | 0.0553 | 0.0495 | 0.0432 | 0.0495 | 0.0460 | 0.0487 | 0.0482 |

**Table 5. Dimensionless super-efficiency slacks of three efficient provinces.**

| Province | Year | Energy | SO2 | Waste Water | Labor | Capital | GDP |
|---|---|---|---|---|---|---|---|
| Beijing | 2008 | 0 | 0.0932 | 0 | 0 | 0 | 0.0865 |
| | 2009 | 0 | 0.0828 | 0 | 0 | 0 | 0.0899 |
| | 2010 | 0.0122 | 0.0862 | 0 | 0 | 0 | 0.0765 |
| | 2011 | 0.0353 | 0.0638 | 0 | 0 | 0 | 0.0618 |
| | 2012 | 0.0325 | 0.0614 | 0 | 0 | 0 | 0.0703 |
| | 2013 | 0.0009 | 0.0640 | 0 | 0 | 0 | 0.0646 |
| | 2014 | 0.0544 | 0.0579 | 0 | 0 | 0 | 0.0530 |
| | 2015 | 0.0550 | 0.0547 | 0 | 0 | 0 | 0.0591 |
| | 2016 | 0.0603 | 0.0293 | 0 | 0 | 0 | 0.0521 |
| | 2017 | 0.0426 | 0.0195 | 0 | 0 | 0 | 0.0765 |
| Chongqiong | 2008 | 0.0023 | 0 | 0 | 0 | 0.0103 | 0 |
| | 2009 | 0 | 0 | 0 | 0 | 0 | 0 |
| | 2010 | 0 | 0 | 0 | 0 | 0 | 0 |
| | 2011 | 0 | 0 | 0 | 0 | 0 | 0 |
| | 2012 | 0 | 0 | 0 | 0 | 0 | 0 |
| | 2013 | 0.0002 | 0 | 0 | 0 | 0 | 0 |
| | 2014 | 0.0019 | 0 | 0 | 0 | 0 | 0 |
| | 2015 | 0.0019 | 0 | 0 | 0 | 0.0058 | 0 |
| | 2016 | 0 | 0 | 0 | 0 | 0.0059 | 0 |
| | 2017 | 0 | 0 | 0 | 0 | 0.0189 | 0 |
| Anhui | 2008 | 0.0065 | 0 | 0 | 0 | 0.0423 | 0 |
| | 2009 | 0 | 0 | 0 | 0 | 0.0478 | 0 |
| | 2010 | 0.0031 | 0 | 0 | 0 | 0.0665 | 0 |
| | 2011 | 0.0075 | 0 | 0 | 0 | 0.0416 | 0 |
| | 2012 | 0 | 0 | 0 | 0 | 0.0644 | 0 |
| | 2013 | 0.0002 | 0 | 0 | 0 | 0.0624 | 0 |
| | 2014 | 0.0092 | 0 | 0 | 0 | 0.0629 | 0 |
| | 2015 | 0.0054 | 0 | 0 | 0 | 0.0464 | 0 |
| | 2016 | 0 | 0 | 0 | 0 | 0.0557 | 0 |
| | 2017 | 0 | 0 | 0.0127 | 0 | 0.0232 | 0 |

three efficient provinces in the eastern, central and western areas, respectively, as examples. Note that in order to eliminate the dimension influence, the slacks mentioned in this subsection have been made dimensionless. The slacks calculated by models (3) and (6) are divided by their corresponding data range in the same period, which is the maximum value of the corresponding item minus the minimum value.

As we can see from Table 5, Beijing is overall efficient due to its lower SO2 emissions and higher GDP from 2008 to 2017 as well as its lower energy consumption from 2010 to 2017; Chongqing is evaluated as overall efficient due to its lower energy consumption in 2008 and from 2013 to 2015, more sufficient capital in 2008 and from 2015 to 2017; the good overall efficiency of Anhui is due to lower energy consumption in 2008, 2010, 2011 and from 2013 to 2015 and lower waste water emissions in 2017 as well as more sufficient capital from 2008 to 2017.

Table 6 shows the dimensionless average super-efficiency slacks of overall efficient provinces. According to Table 6, the excellent efficiency of Beijing and Guangdong is mainly due to lower energy consumption and SO2 emissions, and higher GDP. These two provinces do not

**Table 6. Dimensionless average super-efficiency slacks of efficient provinces.**

| Area | Provinces | Energy | SO2 | Waste water | Labor | Capital | GDP |
|------|-----------|--------|-----|-------------|-------|---------|-----|
| Eastern | Beijing | 0.0293 | 0.0613 | 0 | 0 | 0 | 0.0690 |
| | Tianjin | 0 | 0.0005 | 0.0389 | 0.0321 | 0.0538 | 0 |
| | Shanghai | 0 | 0.0017 | 0 | 0.0168 | 4.58E-05 | 0.0161 |
| | Jiangsu | 0 | 0.0025 | 0 | 0.0478 | 0.2935 | 0.1371 |
| | Shandong | 0 | 0 | 0.1267 | 0 | 0.0359 | 0.0015 |
| | Guangdong | 0.0044 | 0.0743 | 0 | 0 | 0 | 0.1065 |
| | Hainan | 0.0552 | 0.0754 | 0 | 0 | 0.0002 | 0 |
| Central | Anhui | 0.0032 | 0 | 0.0013 | 0 | 0.0513 | 0 |
| | Jiangxi | 0.0223 | 0 | 0 | 0 | 0.0035 | 0 |
| Western | Chongqing | 0.0006 | 0 | 0 | 0 | 0.0041 | 0 |
| | Inner Mongolia | 0 | 0 | 0.0293 | 0 | 0.0102 | 1.77E-05 |
| | Qinghai | 0 | 0 | 0.0192 | 0.0208 | 0 | 0 |

have any advantages in terms of capital, waste water emissions and labor. The good efficiency of Shanghai and Jiangsu comes from more sufficient capital and higher GDP, as well as less labor input and SO2 emissions. Without advantages in energy and GDP, Tianjin is better than other provinces in other four input-output items. Shandong has excellent performance in waste water emissions, capital and GDP, while Hainan has certain advantages in energy, SO2 emissions and capital. Among the five efficient provinces in the central and western areas, Anhui, Jiangxi and Chongqing have similar situations, which all consume less energy and have more sufficient capital. Except that Anhui has a slight advantage in waste water emissions, these three provinces have no advantages in other remaining indicators. Inner Mongolia is efficient due to less waste water emissions and more sufficient capital as well as a slight GDP advantage, while Qinghai is efficient due to less labor and waste water emissions.

To sum up, the seven overall efficient eastern provinces have obvious advantages in SO2 emissions, GDP and capital, no more than two provinces have no advantages in these three items, and more than half of the provinces have no advantages in energy, waste water emissions and labor. The five efficient provinces in the central and western areas have no advantage in terms of SO2 emissions, and most provinces have no advantages in GDP.

Table 7 shows the efficiency slacks of the two provinces with the best and worst overall efficiency ranking among the overall inefficient provinces. Among all these provinces, the overall DUSNM score of Zhejiang ranks 13th, that of Shanxi ranks last. As we see from Table 7, Zhejiang's overall inefficiency is mainly caused by excessive energy consumption, the emissions of SO2 and waste water from 2008 to 2010, excessive labor input in 2009 and 2010, insufficient gross domestic capital formation in 2008, and the excessive gross domestic capital formation in 2017. Therefore, it can be seen that if Zhejiang wants to improve its overall efficiency, it needs to reduce energy consumption, SO2 and waste water emissions from 2008 to 2010, decrease the labor input in 2009 and 2010, increase the gross domestic capital formation in 2008 and reduce the gross domestic capital formation in 2016. With the exception of no excessive waste water emissions in 2008 and 2017 and no insufficient capital in 2013 and 2014, the inputs and outputs of Shanxi in all the rest years are either excessive or insufficient, and the gross domestic capital formation from 2008 to 2017 is insufficient. Thus, if we went to improve Shanxi's overall efficiency, almost every year we need to reduce energy consumption, SO2 and waste water emissions as well as labor, increase capital investment, GDP and gross domestic capital formation.

**Table 7. Dimensionless efficiency slacks of two inefficient provinces.**

| | Year | Energy | SO2 | Waste water | Labor | Capital | GDP | $d_l^{free-,t-1*}$ | $d_l^{free+,t*}$ |
|---|---|---|---|---|---|---|---|---|---|
| Zhejiang | 2008 | 0.0244 | 0.0234 | 0.0057 | 0 | 0 | 0 | 0 | 0.0192 |
| | 2009 | 0.0266 | 0.0594 | 0.0332 | 0.1029 | 0 | 0 | 0 | 0 |
| | 2010 | 0.0371 | 0.0682 | 0.0378 | 0.1184 | 0 | 0 | 0 | 0 |
| | 2011 | 0 | 0 | 0 | 0 | 0 | 0 | 0 | 0 |
| | 2012 | 0 | 0 | 0 | 0 | 0 | 0 | 0 | 0 |
| | 2013 | 0 | 0 | 0 | 0 | 0 | 0 | 0 | 0 |
| | 2014 | 0 | 0 | 0 | 0 | 0 | 0 | 0 | 0 |
| | 2015 | 0 | 0 | 0 | 0 | 0 | 0 | 0 | 0 |
| | 2016 | 0 | 0 | 0 | 0 | 0 | 0 | 0 | 0 |
| | 2017 | 0 | 0 | 0 | 0 | 0 | 0 | 0.0373 | 0 |
| Shanxi | 2008 | 0.3005 | 0.6897 | 0 | 0.1090 | 0.0214 | 0.0910 | 0 | 0.0396 |
| | 2009 | 0.2080 | 0.6233 | 0.0244 | 0.1154 | 0.0392 | 0.0497 | 0 | 0.0335 |
| | 2010 | 0.1957 | 0.6370 | 0.0265 | 0.1143 | 0.0588 | 0.0472 | 0 | 0.0509 |
| | 2011 | 0.2314 | 0.6211 | 0.0195 | 0.1161 | 0.0500 | 0.0489 | 0 | 0.0406 |
| | 2012 | 0.2281 | 0.5993 | 0.0222 | 0.1170 | 0.0189 | 0.0625 | 0 | 0.0361 |
| | 2013 | 0.0062 | 0.6067 | 0.0261 | 0.1180 | 0 | 0.0761 | 0 | 0.0543 |
| | 2014 | 0.3072 | 0.6123 | 0.0253 | 0.1164 | 0.0053 | 0.0944 | 0 | 0.0648 |
| | 2015 | 0.2692 | 0.5976 | 0.0201 | 0.1147 | 0 | 0.1032 | 0 | 0.0807 |
| | 2016 | 0.2634 | 0.5219 | 0.0149 | 0.1179 | 0.0198 | 0.1113 | 0 | 0.0853 |
| | 2017 | 0.2899 | 0.6910 | 0 | 0.1029 | 0.1460 | 0.1151 | 0 | 0.1474 |

Fig 3 shows the annual average inefficient slacks of overall inefficient provinces in the three areas. It can be seen from Subfigure (a) of Fig 3 that the two main reasons for inefficiency of the eastern area are relatively diversified over the years. In 2008, 2009, 2012 and 2014, excessive energy consumption and SO2 emissions are the two main reasons; in 2010, 2011 and 2013, excessive labor input and SO2 emissions become the two main reasons; in 2015–2017, there are too much SO2 emissions and too little capital investment. In addition, the inefficient GDP slack of overall inefficient eastern provinces from 2010 to 2013 is equal to zero, which shows that the GDP of the overall inefficient eastern provinces is sufficient in these years. Similarly, as can be seen from Subfigure (a) of Fig 3, overall inefficient eastern provinces have sufficient capitals in 2008 and from 2012 to 2014, and waste water emissions of the overall inefficient eastern provinces from 2014 to 2017 are also not excessive. All the inputs and outputs of overall inefficient provinces in central and western areas have positive inefficient slacks over the years. Except in 2017, the two major reasons for inefficiency of central and western provinces in sample years are excessive SO2 emissions and labor.

Fig 4 shows the average slack of free links in the three areas over the years. "link+" represents the average of $d_l^{free+,t*}$, "link-" represents the average of $d_l^{free-,t*}$. It is worth mentioning that in 2013, the gross domestic capital formation of eastern provinces is neither insufficient nor excessive. In addition, except that the eastern provinces need to reduce the gross domestic capital formation more than they need to increase in 2011 and 2007, they need to increase gross domestic capital formation more than to decrease it in the remaining eight years. For the central provinces, almost all of them required the increase of gross domestic capital formation from 2007 to 2017. For the western provinces, more gross domestic capital formation needs to be added each year than it needs to be reduced.

In order to improve the overall efficiency of each province, provinces in different areas have different priorities for improvement. Inefficient eastern provinces can mainly focus on

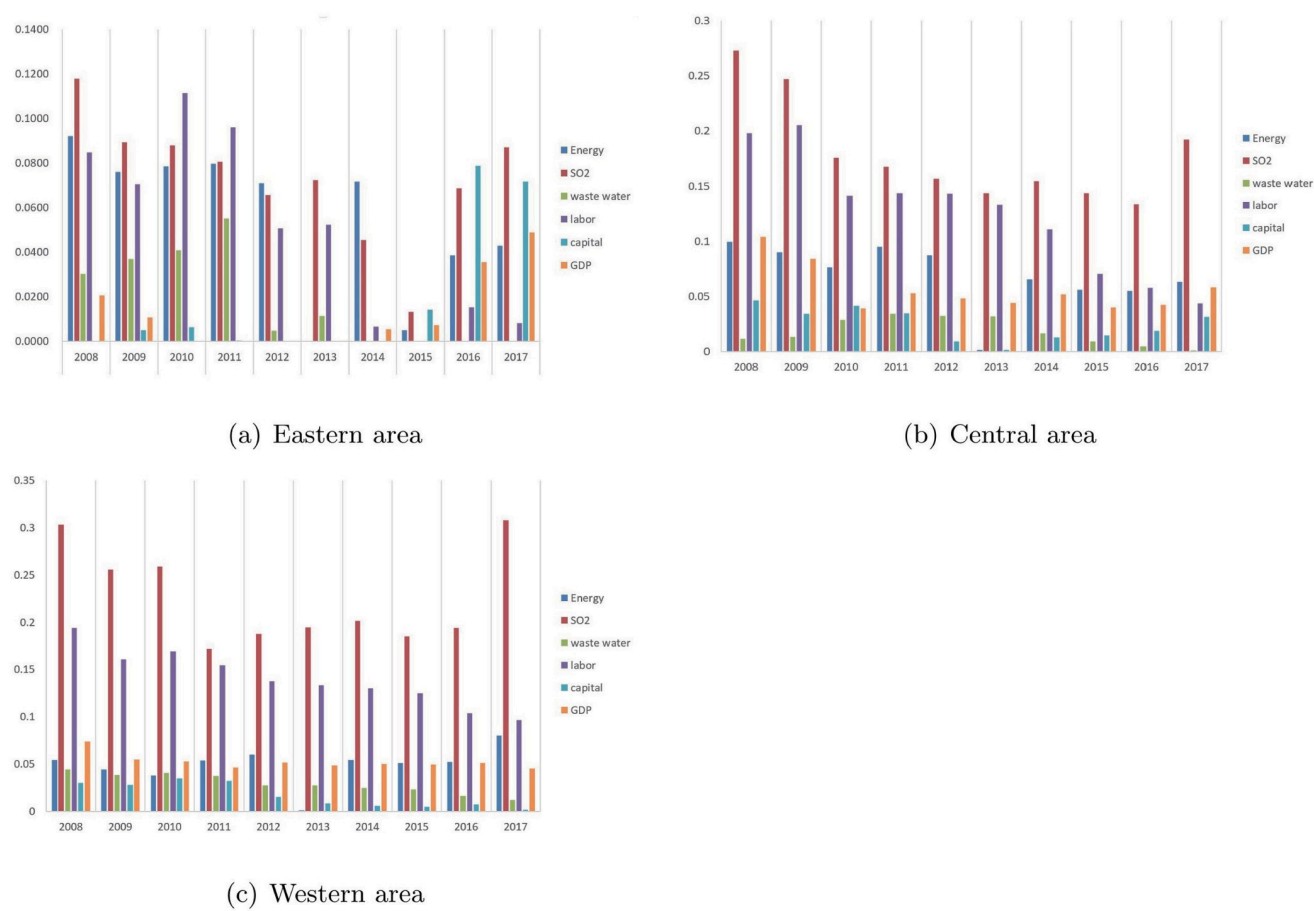

**Fig 3. Annual average inefficiency slacks.** The horizontal axis is the year, and the vertical axis is the average dimensionless slack values of input-output variables. The height of the columns in the bar chart represents the mean dimensionless slacks of the variables with respect to the overall inefficient provinces in each area.

reducing energy consumption, SO2 emissions, and the quantity of labor and increasing capital investment in research and development in the right years. For inefficient eastern provinces, the GDP from 2010 to 2013 and capital in 2008 and from 2012 to 2014 are sufficient and do not have to be increased. Similarly, waste water emissions of the inefficient eastern provinces from 2014 to 2017 have no excesses and do not have to be decreased. For overall inefficient provinces in central and western areas, almost every input and output factor has excess or shortage, and excessive SO2 emissions and labor are two main factors causing the overall inefficiency in most years. Therefore, reducing SO2 emissions and labor can greatly improve the overall efficiency of central and western provinces. They can develop renewable clean energy, advocate the use of clean energy, and reduce the use of non-renewable energy, so as to reduce SO2 emissions. In addition, inefficient provinces in central and western areas need to introduce high-tech automation equipment to reduce labor input. Moreover, in the three areas, gross domestic capital formation of most provinces should be increased in most years.

## Policies

Based on the above performance and slack analysis, we give the following brief policies to improve environmental efficiency of China's provinces.

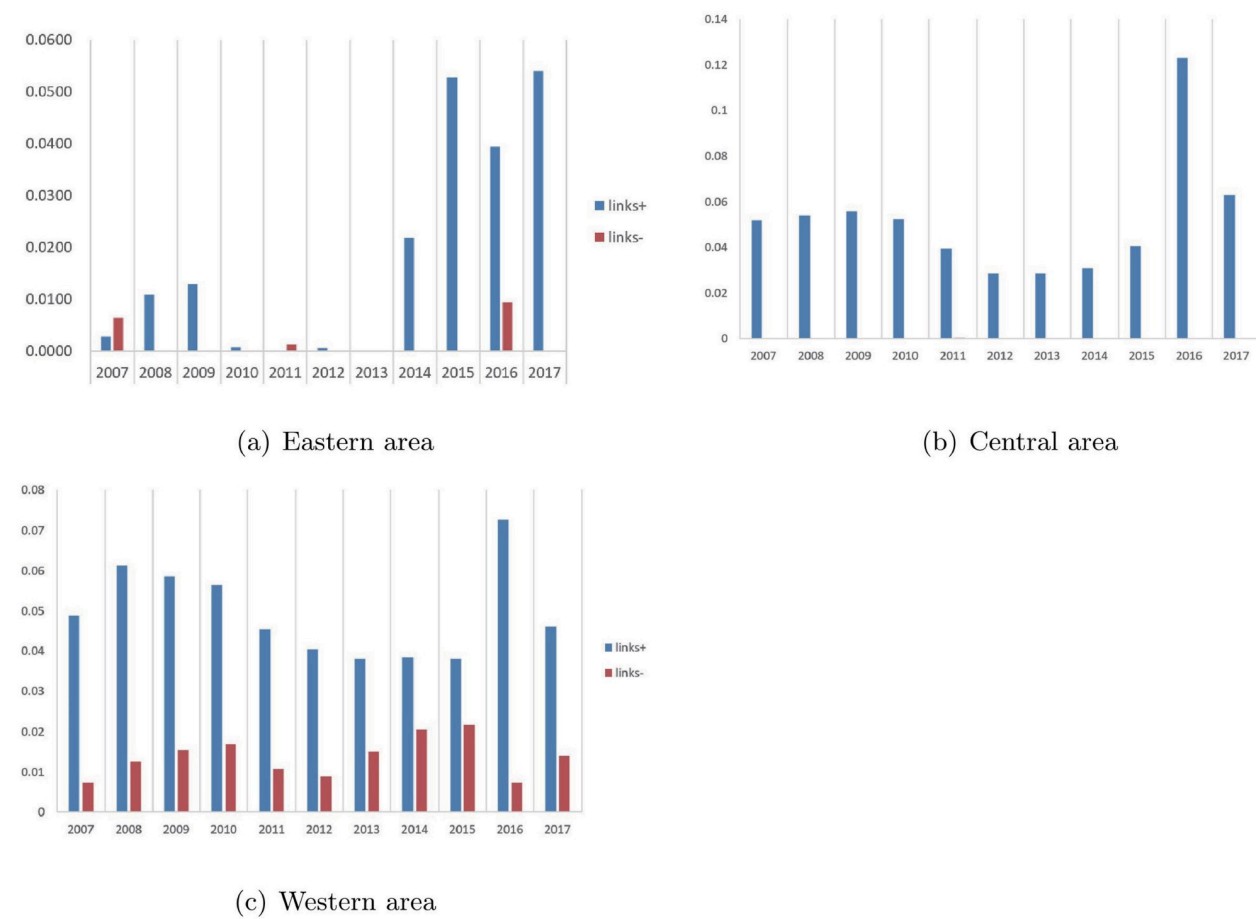

(a) Eastern area

(b) Central area

(c) Western area

**Fig 4. Average slacks of free links.** The horizontal axis is the year, and the vertical axis is the average dimensionless slack values. The height of the columns in the bar chart represents the mean dimensionless $d_l^{free+,t*}$ and $d_l^{free-,t*}$ of the provinces in each region.

1. The industrialization development mode will be changed from resource consumption to technological innovation, and the proportion of major industries should be rationally arranged according to the energy consumption of industries, especially to speed up the adjustment and optimization of the industrial structure in the central and western areas.

2. At the same time of developing economic, promote environmental protection and develop high environmental regulatory standards to reduce the emission of pollutants. This not only needs to improve the environmental awareness of all people, but also to increase the control of pollutant emissions in life and production activities. In addition, it is needed to enhance the cooperation of economic development and environmental protection among individual provinces.

3. The eastern provinces should develop clean energy technology, use renewable clean energy, such as water, wind, solar energy to replace the non-renewable energy that can pollute the environment. The central and western provinces should actively follow the guidance of the central government's policy of supporting development, strengthen technical cooperation with the eastern provinces with good environmental efficiency, increase the efforts to

introduce talents and promote the development of science and technology, improve the innovation ability, so as to achieve economic growth and environmental improvement.

4. When formulating policies on energy conservation and emission reduction, local governments should reduce the impact on economic and social activities, allocate more financial funds to energy conservation, environmental protection and clean energy research, and reduce fossil energy consumption and haze emissions.

## Conclusions and future researches

DEA is a popular method to assess the performance of various organizations in the energy and environment field. Current static environmental efficiency measures, e.g., the UENM, are not suitable for the dynamic system consisting of a sequence of periods. Current dynamic DEA models, e.g., dynamic SBM, do not consider both ND and MD inputs as well as desirable and undesirable outputs. Moreover, UENM and dynamic DEA models cannot further identify the performance of efficient DMUs. To fill the existing research gap, this study proposes the DUENM model and the DUSNM model, and defines the overall DSUNM and period DSUNM scores for DMUs. The overall DSUNM score in the whole dynamic system is a weighted average of period DSUNM scores. As a result, our proposed method can reveal the time when inefficiency or efficiency occurs. According to the overall DUSNM score, we can make a complete ranking for all the DMUs. Moreover, from the slacks generated by our models, we can not only find out the items that cause the inefficiency of an inefficient DMU, but also know in which items an efficient DMU has advantages and in which items an efficient DMU has no advantages.

This study applies the proposed models to a panel data set consisting of 30 provinces in China from 2008 to 2017. From the empirical results, we find that the overall DUSNM score in China shows significant regional differences. Most of the provinces with excellent efficiency are in the eastern area, while most of the provinces with poor performance are in the central and western areas. Slack analysis shows that most efficient provinces in the eastern area have obvious advantages in capital, GDP and SO2 emissions and have no obvious advantages in energy, labor and waste water emissions. Most efficient provinces in the central and western areas have no advantage in terms of SO2 and GDP.

In future research work, we can carry out the following three aspects:

1. From the empirical results, we can see that the efficiencies of provinces in different areas are very different, but our method considers 30 provinces as a reference set, and does not consider the heterogeneity of different areas. The meta-frontier analysis [32] allows taking production technology heterogeneity into consideration. In the meta-frontier analysis, each DMU will be assessed relative to the corresponding group frontier and the meta-frontier, respectively. Then, whether the inefficiency of a DMU is due to its own low management level or the low production technology level of the group can be known. In view of this, it is necessary to explore the DUSNM model in the meta-frontier analysis.

2. Our method does not divide the production process in each period. We can combine dynamic network DEA model [22] to construct DUSNM network model considering production stage division to further explore the efficiency of each production stage.

3. Our method is aimed at the problem of environmental efficiency evaluation. In order to expand the practical application scope of the method, we can also extend it to other performance evaluation application fields by changing the input-output index set.

## Supporting information

**S1 Data.**
(XLSX)

**S2 Data.**
(XLS)

**S1 File.**
(M)

**S2 File.**
(M)

## Author Contributions

**Conceptualization:** Ruiyue Lin.

**Data curation:** Ruiyue Lin.

**Formal analysis:** Ruiyue Lin.

**Funding acquisition:** Ruiyue Lin.

**Investigation:** Ruiyue Lin.

**Project administration:** Zongxin Li.

**Resources:** Zongxin Li.

**Software:** Zongxin Li.

**Supervision:** Zongxin Li.

**Validation:** Zongxin Li.

**Visualization:** Zongxin Li.

**Writing – original draft:** Ruiyue Lin.

**Writing – review & editing:** Ruiyue Lin.

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
