## [Decision Letter · Decision Letter 0]

21 Jun 2023

PONE-D-23-17252Intertemporal environmental performance assessment in China: a new network-based dynamic super-efficiency measurePLOS ONE

Dear Dr. Lin,

Thank you for submitting your manuscript to PLOS ONE. After careful consideration, we feel that it has merit but does not fully meet PLOS ONE’s publication criteria as it currently stands. Therefore, we invite you to submit a revised version of the manuscript that addresses the points raised during the review process.

We look forward to receiving your revised manuscript.

Kind regards,

Qunxi Gong

Academic Editor

PLOS ONE

Journal Requirements:

   "The work was supported by the National Natural Science Foundation of China [grant number 71971163]."

   "Ruiyue Lin recieved the National Natural Science Foundation of China 

[grant number 71971163]. URL: https://grants.nsfc.gov.cn/egrantindex/funcindex/prjsearch-list?locale=zh_CN.  The funders had no role in study design, data collection and analysis, decision to publish, or preparation of the manuscript.

Other authors received no specific funding for this work."

Reviewers' comments:

Reviewer's Responses to Questions

**Comments to the Author**

1. Is the manuscript technically sound, and do the data support the conclusions?

Reviewer #1: Yes

Reviewer #2: Yes

Reviewer #3: Yes

2. Has the statistical analysis been performed appropriately and rigorously? 

Reviewer #1: Yes

Reviewer #2: Yes

Reviewer #3: Yes

3. Have the authors made all data underlying the findings in their manuscript fully available?

Reviewer #1: Yes

Reviewer #2: Yes

Reviewer #3: Yes

4. Is the manuscript presented in an intelligible fashion and written in standard English?

Reviewer #1: Yes

Reviewer #2: Yes

Reviewer #3: Yes

5. Review Comments to the Author

Reviewer #1: Title: Intertemporal environmental performance assessment in China: a new network-based dynamic super-efficiency measure This paper proposes a dynamic DEA model and a dynamic super-efficiency DEA model for the system consisting of a sequence of periods linked by carryovers. Compared to previous studies, the proposed dynamic DEA measure considers desirable and undesirable outputs, inputs under both natural disposability and managerial disposability, can completely rank the overall performance of all the decision-making units (DMUs) and provide information about when and what factors lead to inefficiency or efficiency of DMUs. The proposed models are applied to examine the environmental performance of 30 provinces in China during the 2008-2017 period. My detailed comments are as follows

Abstract: The abstract should sequence the objectives, techniques, and concise results. Revise accordingly. Check the sequence of keywords.

Introduction: I can see that the introduction section is lengthy. Exclude unnecessary information and be concise with the study’s research problem, gaps, objectives, and innovative contribution if possible. Usually, at the end of the introduction section, the authors describe the innovations and route of the study. Revise accordingly to put the introduction section in the appropriate shape.

Methods: Proposed methods are accurate and innovative

Add more citations in the interpretation of the result section. Further, elaborate on the results in detail.

Conclusions and policy implications. Add limitations and future research ideas in the conclusion section. Give more practical implications of your proposed model.

Avoid grammatical and typo errors, and revise the manuscripts for these concerns.

Carefully check and revise the table and figure numbers in the manuscript.

Reviewer #2: Title: Intertemporal environmental performance assessment in China: a new network-based dynamic super-efficiency Measure"

Manuscript Number: PONE-D-23-17252

I have completed the review of the manuscript "Dynamic Data Envelopment Analysis for Measuring Intertemporal Environmental Efficiency: Evidence from Chinese Provinces." The authors present a novel approach using a dynamic data envelopment analysis (DEA) model to assess the intertemporal environmental efficiency of decision-making units (DMUs) in a dynamic setting. The manuscript investigates the environmental performance of 30 provinces in China over the period 2008-2017 and provides insightful findings on regional differences and factors contributing to inefficiency. Overall, the study is valuable and contributes to the literature on environmental efficiency measurement. However, I recommend minor revisions to improve clarity and strengthen the analysis.

The authors should enhance the clarity and organization of the manuscript by providing a more straightforward introduction that clearly states the research objectives, the significance of the research question, and the methodology employed. Additionally, it would be helpful to provide a brief overview of the structure of the paper to guide readers through the sections.

The manuscript would benefit from a more detailed explanation of the network-based dynamic DEA model and the dynamic super-efficiency DEA model proposed by the authors. Including the mathematical formulations, constraints, and variables used in these models would assist readers in understanding the methodology and replicating the analysis.

Provide a comprehensive description of the data sources used in the study, including information on the variables used to measure environmental efficiency and regional differences. Additionally, it would be beneficial to clarify how the DMUs were selected and if any data preprocessing techniques were applied.

The authors should present the results of the dynamic DEA and super-efficiency DEA models. The findings related to the regional differences in environmental performance should be summarized and discussed concisely and coherently. Provide clear explanations of the implications and policy recommendations that arise from the results.

The manuscript would benefit from a discussion of the limitations of the proposed models and potential avenues for future research. Addressing these aspects would enhance the paper's contribution and help readers understand the broader context of the study.

Give a section number to each section

Revise for typo mistakes

Reviewer #3: The design of the paper adopts the dynamic DEA model in the research method, and the method is relatively conventional. 10 years of data were used to analyze the environmental performance of 30 provinces in Chinese mainland. The following comments need to be addressed:

1.The indicators selected should be supplemented by explanations as to why they were chosen for empirical research

2. References to policy recommendations or management decisions are overly simplistic, and in-depth discussions are recommended.

3. Please strengthen the discussion of differences between similar studies and those studied, highlighting the contribution of this study.

4. Please add the limitations and prospects of this study.

6. PLOS authors have the option to publish the peer review history of their article (what does this mean?). If published, this will include your full peer review and any attached files.

Reviewer #1: No

Reviewer #2: No

Reviewer #3: No

---

## [Author Response · Author response to Decision Letter 0]

20 Jul 2023

We’d like to express our heartfelt thanks to the editor for giving us an opportunity to revise the manuscript. We do appreciate it very much!

During the past several weeks, we have modified our manuscript carefully according to the reviewers’reports. All the comments raised by the two reviewers are answered. What follows are our detailed responses. To distinguish, we put their original comments in Italic in the following. Please see the following point-by-point responses.

Editor’s Comments:

1 . Please ensure that your manuscript meets PLOS ONE’s style requirements, including those for file naming. The PLOS ONE style templates can be found at ...

Answer: We have downloaded the latex template of plos from the website, and have applied the article format according to the template requirements. We hope the present format of the manuscript can meet PLOS ONE’s style requirements.

2. Thank you for stating the following in the Acknowledgments Section of your manuscript: ”The work was supported by the National Natural Science Foundation of China [grant num- ber 71971163]. ”

We note that you have provided funding information that is not currently declared in your Funding Statement. However, funding information should not appear in the Acknowledg- ments section or other areas of your manuscript. We will only publish funding information present in the Funding Statement section of the online submission form. Please remove any funding-related text from the manuscript and let us know how you would like to update your Funding Statement. ... Other authors received no specific funding for this work. ”

Answer: We have deleted the funding information from the Acknowledgments section of the revised manuscript.

3. We note that Figure 1 in your submission contain [map/satellite] images which may be copyrighted. ... We require you to either (a) present written permission from the copyright holder to publish these figures specifically under the CC BY 4 . 0 license, or (b) remove the figures from your submission

Answer: Figure 1 in the original manuscript is our original drawing and does not involve any copyright issues. We don’t know why we have this issue. In order to comply with the journal’s request, we removed this figure in the revised manuscript. We hope that this change will not affect readers’ reading of this article. Please see the modification on page 4.

4 . Please include captions for your Supporting Information files at the end of your manuscript, and update any in-text citations to match accordingly.

Answer: We have made changes in the revised manuscript. We hope that this revision can meet the requirements of the journal.

5. Please review your reference list to ensure that it is complete and correct. ...If you need to cite a retracted article, indicate the article’s retracted status in the References list and also include a citation and full reference for the retraction notice .

Answer: We have checked our reference list. We ensure that it is complete and correct. All the comments raised by the three reviewers are answered. Please see

the following point-by-point responses.

Reviewer 1’s Comments:

1 Abstract: The abstract should sequence the objectives, techniques, and concise results . Re- vise accordingly. Check the sequence of keywords .

Answer: We have revised the abstract according to this comment and the sequence of keywords has also been adjusted. Please see the revised abstract and keywords. 2 Introduction: I can see that the introduction section is lengthy. Exclude unnecessary infor- mation and be concise with the study’s research problem, gaps, objectives, and innovative contribution if possible . Usually, at the end of the introduction section, the authors describe the innovations and route of the study. Revise accordingly to put the introduction section in the appropriate shape .

Answer: According to this comment, we greatly reduced the content of the introduction section, which briefly introduces the research problems, gaps, objectives, and innovative

contribution. Please see the revised introduction section from page 2 to page 3.

After combining the opinions of other reviewers, the penultimate paragraph of intro- duction describes the main innovations and routes of the study, and the last paragraph describes the structure of the article. Please see the last two paragraphs of the introduction section on page 3.

3 Methods: Proposed methods are accurate and innovative . Add more citations in the inter- pretation of the result section. Further, elaborate on the results in detail.

Answer: Combining this opinion with reviewer 2’s comments, we significantly modified Section 2 and Section 3, mainly adding the explanation of the model. In the part of empirical results, we mainly added explanations for the selection of indicators, relevant literature citations, and comparisons with existing methods, deleted some unnecessary descriptions, and carried out further analysis of empirical results. I hope the current modification can meet the reviewers’requirements.

4 Conclusions and policy implications . Add limitations and future research ideas in the con- clusion section. Give more practical implications of your proposed model.

Answer: We have added limitations and future research ideas in the revised conclusion section. More practical implications of the proposed model are also added in the first paragraph of the revised conclusion section. Please see the modification from page 17 to page 18.

5 Avoid grammatical and typo errors, and revise the manuscripts for these concerns . Answer: We have carefully reviewed the manuscript many times and have corrected

many grammatical and typos. I hope the revised manuscript can meet the reviewers’re- quirements.

6 Carefully check and revise the table and figure numbers in the manuscript.

Answer: In the revised manuscript, we have deleted 2 figures, and now there are 4 figures and 7 tables. We also corrected the quotes in the revised manuscript.

Reviewer 2’s Comments:

1 . The authors should enhance the clarity and organization of the manuscript by providing a more straightforward introduction that clearly states the research objectives, the significance of the research question, and the methodology employed. Additionally, it would be helpful to provide a brief overview of the structure of the paper to guide readers through the sections .

Answer: We have greatly reduced the content of the introduction section, which briefly introduced the research problems, gaps, objectives, and innovative contribution. The last paragraph of the introduction section has described the structure of the article. Please see the revised introduction section from page 2 to page 3.

2. The manuscript would benefit from a more detailed explanation of the network- based dynamic DEA model and the dynamic super- efficiency DEA model proposed by the authors . Including the mathematical formulations, constraints, and variables used in these models would assist readers in understanding the methodology and replicating the analysis .

Answer: Thanks for this comment! Although we also have explanations of variables and constraints in the original manuscript, the layout is messy. In the revised manuscript, we concentrated the description of the variables and constraints at the bottom of the models. Please see the modification from page 4 to page 9.

3. Provide a comprehensive description of the data sources used in the study, including information on the variables used to measure environmental efficiency and regional differ- ences . Additionally, it would be beneficial to clarify how the DMUs were selected and if any data preprocessing techniques were applied.

Answer: According to this comment, we revised Table 2 to describe the data source. Please see the revised Table 2 on page 11.

In addition, it should be noted that we did not specifically select indicators to measure regional differences. We only selected capital, labor, and energy as inputs, polluting emis- sions as undesirable outputs, and GDP as desirable outputs according to the literature on environmental efficiency evaluation. It’s just that in reality, the differences between different regions do exist, rather than the variables we chose to measure the differences.

Since we measure inter-provincial environmental efficiency, we definitely choose provinces as DMUs. Since DEA is a dimensionless technology, all the data are directly substituted into the models to calculate without data processing. Please see our description on pages 10 and 11.

4 . The authors should present the results of the dynamic DEA and super- efficiency DEA models .

Answer: According to our method, the period efficiency scores are derived from these two models, so in order to avoid duplication, we only add the overall efficiency scores generated by these two models in Table 3. See the changes in Table 3 and the related

modification from page 11 to page 12.

5. The findings related to the regional differences in environmental performance should be summarized and discussed concisely and coherently. Provide clear explanations of the implications and policy recommendations that arise from the results .

Answer: Based on this comment and combined with the comments of other reviewers, we have significantly reduced the description of regional differences. The explanations of the results has been revised accordingly and a subsection has been added to present the policies. Please see the modification on pages 12, 13 and 17.

6. The manuscript would benefit from a discussion of the limitations of the proposed mod- els and potential avenues for future research. Addressing these aspects would enhance the paper’s contribution and help readers understand the broader context of the study.

Answer: Thanks for this comment! Please see our modification on page 18.

7. Give a section number to each section

Answer: The template for Plos one requires no numbers in each section.

8. Revise for typo mistakes

Answer: We have carefully reviewed the manuscript many times and have corrected many typos. I hope the revised manuscript can meet the requirements.

Reviewer 3’s Comments:

1 . The indicators selected should be supplemented by explanations as to why they were chosen for empirical research.

Answer: Thanks for this important comment! We have added the explanations about why the indicators are chosen in the revised manuscript. Please see the modification on page 11.

2. References to policy recommendations or management decisions are overly simplistic, and in- depth discussions are recommended.

Answer: According to this comment, we have added an subsection about policy rec- ommendations. Please see the modification on page 17. And some in-depth discussions are provided on pages 15-17.

3. Please strengthen the discussion of differences between similar studies and those studied, highlighting the contribution of this study.

Answer: According to this comment, we do two main modifications:

(1) We highlight the contributions of this study on page 3.

(2) In empirical research, we compare the results generated by our method with those generated by similar studies. Please see the modification from page 11 to page 12.

4 . Please add the limitations and prospects of this study.

Answer: We have added the descriptions about limitations and prospects of this study on page 17.

We do hope that our thorough modifications and the above detailed illustrations are good enough to fully satisfy reviewers’ and the editor’s requirements, and our revised paper can be accepted to publish in Plos one!

We are looking forward to your positive response!

For more detailed responses, please see Response to Reviewers in the Attach Files.

---

## [Decision Letter · Decision Letter 1]

18 Aug 2023

Intertemporal environmental efficiency assessment in China: a new network-based dynamic super-efficiency measure

PONE-D-23-17252R1

Dear Dr. Lin,

We’re pleased to inform you that your manuscript has been judged scientifically suitable for publication and will be formally accepted for publication once it meets all outstanding technical requirements.

Kind regards,

Qunxi Gong

Academic Editor

PLOS ONE

Additional Editor Comments (optional):

Reviewers' comments:

Reviewer's Responses to Questions

**Comments to the Author**

1. If the authors have adequately addressed your comments raised in a previous round of review and you feel that this manuscript is now acceptable for publication, you may indicate that here to bypass the “Comments to the Author” section, enter your conflict of interest statement in the “Confidential to Editor” section, and submit your "Accept" recommendation.

Reviewer #1: All comments have been addressed

Reviewer #3: All comments have been addressed

2. Is the manuscript technically sound, and do the data support the conclusions?

Reviewer #1: Yes

Reviewer #3: Yes

3. Has the statistical analysis been performed appropriately and rigorously? 

Reviewer #1: Yes

Reviewer #3: Yes

4. Have the authors made all data underlying the findings in their manuscript fully available?

Reviewer #1: Yes

Reviewer #3: Yes

5. Is the manuscript presented in an intelligible fashion and written in standard English?

Reviewer #1: No

Reviewer #3: Yes

6. Review Comments to the Author

Reviewer #1: We can see that all comments addressed. Authors don no need to further do any modifications in this manuscript.

Reviewer #3: This edition of the paper has revised well as I understand, and my comments have been addressed. No more suggestions so far.

7. PLOS authors have the option to publish the peer review history of their article (what does this mean?). If published, this will include your full peer review and any attached files.

Reviewer #1: No

Reviewer #3: **Yes: **Li Ying

---

## [Editor Report · Acceptance letter]

22 Aug 2023

PONE-D-23-17252R1 

Intertemporal environmental efficiency assessment in China: a new network-based dynamic super-efficiency measure 

Dear Dr. Lin:

I'm pleased to inform you that your manuscript has been deemed suitable for publication in PLOS ONE. Congratulations! Your manuscript is now with our production department. 

Kind regards, 

on behalf of

Dr. Qunxi Gong 

Academic Editor

PLOS ONE